# Preparation and Application of Molecularly Imprinted Polymers for Flavonoids: Review and Perspective

**DOI:** 10.3390/molecules27217355

**Published:** 2022-10-29

**Authors:** Yurou Yang, Xiantao Shen

**Affiliations:** State Key Laboratory of Environment Health (Incubation), Key Laboratory of Environment and Health, Ministry of Education, Key Laboratory of Environment and Health (Wuhan), Ministry of Environmental Protection, School of Public Health, Tongji Medical College, Huazhong University of Science and Technology, Hangkong Road #13, Wuhan 430030, China

**Keywords:** natural products, flavonoids, molecularly imprinted polymers, preparation methods, applications

## Abstract

The separation and detection of flavonoids from various natural products have attracted increasing attention in the field of natural product research and development. Depending on the high specificity of molecularly imprinted polymers (MIPs), MIPs are proposed as efficient adsorbents for the selective extraction and separation of flavonoids from complex samples. At present, a comprehensive review article to summarize the separation and purification of flavonoids using molecular imprinting, and the employment of MIP-based sensors for the detection of flavonoids is still lacking. Here, we reviewed the general preparation methods of MIPs towards flavonoids, including bulk polymerization, precipitation polymerization, surface imprinting and emulsion polymerization. Additionally, a variety of applications of MIPs towards flavonoids are summarized, such as the different forms of MIP-based solid phase extraction (SPE) for the separation of flavonoids, and the MIP-based sensors for the detection of flavonoids. Finally, we discussed the advantages and disadvantages of the current synthetic methods for preparing MIPs of flavonoids and prospected the approaches for detecting flavonoids in the future. The purpose of this review is to provide helpful suggestions for the novel preparation methods of MIPs for the extraction of flavonoids and emerging applications of MIPs for the detection of flavonoids from natural products and biological samples.

## 1. Introduction

Flavonoids, which are a class of low-molecular-weight substances synthesized from *L*-phenylalanine in plants, have been widely found in fruits, vegetables, traditional Chinese medicine and their related products (such as chocolate, coffee, wine and tea) [1,2,3]. Flavonoids are a giant crowd of structurally related compounds with a chroman-type skeleton (basic structures composed of 15 carbon atoms). According to the substitutions in the groups, flavonoids can be divided into several classes, mainly including flavonols, flavanones, flavones, isoflavonoids, flavanonols and chalcones (Figure 1) [4,5,6]. Due to their excellent properties (antioxidant, hypoglycemic, anti-tumor and anti-bacterial activities), flavonoids have been widely used in the treatment of numerous diseases such as cancers, diabetes, osteoarthritis, ocular disorders and cardiovascular diseases [1,5,6,7,8,9,10]. Because of the important applications shown above, it is essential to analyze the levels of flavonoids in natural products. Usually, before the detection, the flavonoids in the samples need to be selectively extracted and separated. At present, the commonly used extraction method for flavonoid separation is solid phase extraction (SPE). However, the traditional SPE method often lacks selectivity, resulting in low detection sensitivity. Therefore, to enhance the analysis sensitivity for flavonoids in complex samples, new adsorbents with high specificity are needed for the SPE column.

Molecular imprinting is a synthetic method for the generation of receptors, also named molecularly imprinted polymers (MIPs), which have tailor-made recognition sites towards targets [11]. In general, MIPs are usually used as adsorbents of SPE, monolithic columns for chromatographic separation and simulative enzymes for catalysis. In addition, they are widely used in the field of clinical drug analysis and biomimetic sensing [12]. Recently, it has been used as artificial biomimetic material, such as artificial antibodies (plastibodies), enzymes and other biological molecules [13,14,15,16]. The basic principle of molecular imprinting is mainly based on Pauling’s theory of bioimprinting and the lock-and-key concept [17,18,19,20]. Figure 2 shows the typical steps for synthesizing MIPs. Generally, MIPs possess the characteristics of having a low cost, simple preparation and high physical stability. In consequence, MIPs have received increasing attention in various fields [21]. Because of the above advantages, MIPs have also been applied in the selective separation of flavonoids. For instance, MIP particles as adsorbents for SPE [12,22,23,24,25] and molecularly imprinted membranes (MIMs)-based separation have been extensively reported in the literature. Furthermore, MIPs can be successfully used as the recognition elements of a sensor to detect flavonoids in samples. However, to the best of our knowledge, a comprehensive review article to address the applications of molecular imprinting in flavonoid separation and purification of flavonoids is still lacking.

In this paper, we reviewed the progress of MIPs in the separation, purification and detection of flavonoids in the past five years. The first part of this review discussed the common strategies for preparing MIPs towards flavonoids, and the second part of this review summarized the methods for the separation, purification and detection of flavonoids by MIPs. Last, we discussed the advantages and disadvantages of the current synthetic methods for preparing MIPs of flavonoids and prospected the approaches for detecting flavonoids in the future.

## 2. Methods for Preparing MIPs of Target Flavonoids

Molecular imprinting can be generally divided into covalent, semi-covalent and non-covalent methods according to the binding modes of different sites of action (interaction between the templates and the functional groups of the monomers) [26,27]. In the process of preparing MIPs, it is generally necessary to consider the selection and proportion of template molecules, functional monomers, cross-linker, initiators and porogen, etc. In the literature, MIPs towards flavonoids have been successfully prepared by bulk polymerization, precipitation polymerization, surface imprinting and emulsion polymerization.

### 2.1. Bulk Polymerization

Originally, the simplest approach to synthesizing MIPs is bulk polymerization. A typical synthesis process of MIPs by bulk polymerization contained two main steps: radical polymerization, and mechanical grinding and sieving [25,26,27]. Typically, the templates and functional monomers are mixed in the solvent for pre-polymerization. After the addition of a cross-linker, initiators and porogen to the mixture, polymerization of the system is conducted. The bulk polymers are mechanically ground and sieved. After elution of the template from the particles, the MIPs are finally obtained.

MIPs of flavonoids have also been successfully prepared by bulk polymerization. For example, Xie et al. [28] synthesized calycosin-MIPs using bulk polymerization resoundingly which could selectively identify flavonoid aglycons in Astragali Radix. At the same time, Ersoy et al. [29] synthesized quercetin-imprinted polymer by bulk polymerization using 4-vinylphenylboronic acid (4-VP) as the functional monomer. To investigate the property of this monomer, Huang et al. [30] prepared different types of MIPs towards quercetin using 4-VP and acrylamide (AA) as functional monomers, respectively. The data showed that the maximum equilibrium adsorption capacities of MIP-4VP and MIP-AM for quercetin were 0.397 mg/g and 0.283 mg/g, respectively. However, the above MIPs were prepared using only a single template. Recently, MIPs towards flavonoids with dual templates have also been developed in some literature. For example, Zhang et al. [31] prepared double-template MIPs (dMIPs) towards both quercetin and schisandrin b via bulk polymerization. The dMIPs were spherical and closely connected to each other, forming plate-like structures with gullies on the surface. The maximum adsorption capacity of quercetin by dMIPs was 23.6 mg/g.

Although bulk polymerization for the synthesis of flavonoids MIPs is simple, it still faces some disadvantages. The biggest disadvantage of this method is that the obtained MIPs need mechanical grinding, and the procedure is time-consuming and laborious. Moreover, the shape of MIPs prepared by bulk polymerization is irregular, which extensively limits their applications in flavonoid separation. In order to solve these problems, new polymerization approaches have been presented recently [25,32].

### 2.2. Precipitation Polymerization

It is known in polymer chemistry that, in a typical precipitation polymerization, the monomers, cross-linkers and initiators used in the preparation process are soluble in the solvent, while the generated polymer is insoluble in the reaction system, so the polymers precipitate out of the reaction solution [26,33]. The steps of precipitation polymerization are roughly the same as those of traditional polymerization methods. However, the reaction system of precipitation polymerization is simpler in several aspects: there is no need for surfactant and other solvents, there is less chance of losing the functional monomer, and it is much easier to control the polymer particle size [34,35]. Recently, the synthesis of MIPs using precipitation polymerization has been widely reported [36,37]. In a typical synthesis, the imprinting system is dissolved in the corresponding solvent, and the polymers insoluble in the reaction system are obtained after polymerization reaction, and then the polymers are collected by centrifugation or filtration, etc. After elution of the template by appropriate solvent, the MIPs are finally obtained by cleaning and drying.

Recently, large numbers of surveys have shown that various MIPs towards flavonoids could be prepared by precipitation polymerization [26,37]. For instance, MIPs towards myricetin were successfully prepared using precipitation polymerization by Wan et al. [37]. The MIPs achieved separation of the target myricetin from practical herbal medicines samples.

For the traditional MIPs synthesized by precipitation polymerization, they usually show a much lower surface area, which limits the binding capacity of the MIPs towards the target flavonoids. To enhance the adsorption capacity of the MIPs, Abdullah et al. [38] synthesized magnetic MIPs using quercetin as templates by a modified precipitation polymerization method. The prepared MIPs were in the form of honeycomb particles and had a much high adsorption capacity (85.5 mg/g) towards quercetin. In addition, it was successfully used as an adsorbent for the selective extraction of quercetin from the onion. Modification of the imprinting system can also increase the binding efficiency of the MIPs towards the target flavonoids. Recently, Chrzanowska et al. [39] reported that the binding selectivity of the MIPs synthesized using 2-(dimethylamino) ethyl methacrylate (DEM) as monomers was 5 folds higher than that synthesized using methacrylic acid (MAA) as monomers. Li et al. [40] reported that fabrication of binary MIPs (Bi-MIPs) using MAA and AM as binary functional monomers was also an efficient way to enhance the adsorption performance of the MIPs (the imprinting factor (IF) of the Bi-MIPs was 1.54 times to the ordinary MIPs).

In brief, all the above articles have prepared plentiful kinds of MIPs towards flavonoids by precipitation polymerization. Generally, the MIPs synthesized by precipitation polymerization are microspheres with narrow dispersion, regular shape, uneven size and smooth surface, which provides the MIPs with promising applications in sensing (the uniform particles showed uniform signal for analysis). However, the reaction system of precipitation polymerization is carried out in diluted solvents, which requires a large number of organic solvents. Obviously, this does not satisfy the principle of green chemistry.

### 2.3. Surface Imprinting

Compared with the methods proposed above, surface imprinting is a facile, fast, convenient and straightforward method for synthesis of MIPs towards flavonoids [26,41]. This is due to the fact that surface imprinting can address the problem of template elution in molecular imprinting [41,42]. In the literature, the preparation of MIPs by surface imprinting towards flavonoids including flavonols, flavanones, flavones, isoflavones, flavanonols and chalcones has been briefly described.

Flavonols (e.g., quercetin, myricetin, rutin, morin and kaempferol) are the most common group of flavonoids. The MIPs towards quercetin had been reported [43]. For example, to extract the quercetin from apple samples, Cheng et al. [44] synthesized Fe_3_O_4_@MIPs using surface imprinting, and the maximum adsorption capacity of quercetin by the MIPs was 10.5 mg/g. Similarly, preparation of quercetin MIPs by surface imprinting has also been reported for extraction of quercetin from other substances including Ginkgo Biloba [45], Ginkgo Biloba extract capsule [46], onion epidermis [47], human urine [47] and rat plasma [48]. MIPs towards other flavonols (besides quercetin) by surface imprinting were also reported [44,45,46,47,48,49,50,51,52]. For example, Zhang et al. [49] prepared novel imprinted quantum dots towards myricetin based on boronate affinity-based template-immobilization surface imprinting, and the IF was evaluated to 7.88. Song et al. [50] synthesized MIPs towards rutin with a thick and smooth surface, which could concentrate rutin from Sophora Japonica. To isolate morin from Sanghuangporus Lonicericola, Zhang et al. [51] synthesized morin magnetic MIP nanoparticles using morin as templates by surface imprinting. The maximum adsorption capacity of MIPs towards morin was 3.24 mg/g, which was 2.55 times that of NIPs (1.27 mg/g). Recently, Cheng et al. [52] and Ma et al. [53] both reported the synthesis of MIPs for the extraction of kaempferol from apples and Rhododendron species.

Naringin and hesperidin are both flavanones. Successful preparation of MIPs of naringin [54,55] and hesperidin [56] has been reported. To selectively separate naringin from Citri Grandis, Pan et al. [55] prepared MIP microspheres using surface imprinting, the IF of the MIP microspheres was 2.89. Using the MIPs as molecular recognition elements, a fluorescent magnetic MIP sensor towards naringin was successfully constructed [54]. The detection limit (LOD) of the sensing method was 0.100 mg/L. Furthermore, Wang et al. [56] prepared magnetic MIPs towards hesperidin. The maximum adsorption capacity of magnetic MIPs (16.6 mg/g) to the target analyte was much higher than that of magnetic NIPs (7.55 mg/g).

Besides flavonols and flavanones, other flavonoids including luteolin [57,58], formononetin [59], phloridzin [60] and silybin [61] were also successfully used as templates to synthesize MIPs. For example, luteolin is one of the crucial flavones. Recently, MIPs of luteolin had been triumphantly synthesized by Lu et al. [57] using electropolymerization. The MIP composite membranes synthesized by surface imprinting possessed a cauliflower structure. It is known that electropolymerization is one of indispensable methods of surface imprinting. In this method, it is no need for any initiators. Due to this merit, electropolymerization has been a useful polymerization method for molecular imprinting. Furthermore, Wei et al. [58] also using electropolymerization synthesized MIP membranes with the features of roughness and multiple pores for detecting luteolin. Besides MIP membranes, MIP particles with unique core–shell structures were also successfully reported by surface imprinting [59,60]. Zhang et al. [59] and Gao et al. [60] both prepared MIPs with unique core–shell structures by surface imprinting. The former uses formononetin as the template, *N*-isopropylacrylamide (NiPAm) as the thermosensitive functional monomer, MAA as the auxiliary functional monomer, and preparation of adsorption capacity of 16.4 mg/g using magnetic dual-responsive MIPs. The latter synthesized magnetic MIPs using phloridzin as templates, Fe_3_O_4_@SiO_2_@NH_2_ (self-made) as functional monomer and the equilibrium adsorption capacity of the polymer at 60 min (14.6 mg/g) was much higher than that of the NIPs (4.40 mg/g).

As described in this subsection, surface imprinting is one of the most common methods for the synthesis of MIPs towards flavonoids. So far, MIPs by surface imprinting have been successful for separation of flavonols, flavanones, flavones, isoflavones, flavanonols, chalcones. However, the preparation of virtual template MIPs towards flavonoids by surface imprinting has not been reported at the present.

### 2.4. Emulsion Polymerization

Emulsion polymerization, which is easy to heat dissipation and achieve serialization, is one of the most versatile and indispensable methods for polymer synthesis [62]. The main principle of emulsion polymerization is the polymerization of dispersed monomers in a continuous phase in the presence of surfactant [63,64]. Recently, emulsion polymerization has been successfully used to prepare MIPs towards flavonoids [65,66,67]. For example, by using Triton X-100 as the surfactant, Xu et al. [65] prepared fluorescent MIPs for the determination of quercetin in grape juice, tea juice, black tea and red wine by emulsion polymerization.

Small solid particles can also be used as emulators to stabilize an oil/water or water/oil emulsion. The resulting emulsion was named Pickering emulsion. Pickering emulsion polymerization is one of the emerging and significant emulsion polymerizations [68]. So far, Pickering emulsion polymerization has been successfully applied in preparation of MIPs [69,70,71,72], including the production of MIPs towards flavonoids. For example, by using modified hydroxyapatite (whose external surface was modified hydrophobic groups) acted as a solid surfactant, Sun et al. [73] synthesized MIP microspheres via Pickering emulsion polymerization for extraction of quercetin in Spina Gleditsiae.

As a summary, the main parameters in preparation of MIPs towards flavonoids were listed in Table 1. It was seen that the traditional functional monomers and various materials had been used as functional monomers in this field. Briefly, the reported traditional functional monomers included AA, MAA, 4-VP, 2- vinylphenylboronic acid (2-VP). Moreover, some novel materials were also used as functional monomers such as self-made materials (Fe_3_O_4_@SiO_2_@NH_2_, and MC*-modified Fe_3_O_4_) and new materials (cyclodextrin and deep eutectic solvents). Another important thing to mention here was the initiation way of the polymerization. In general, the polymerization initiation methods of MIPs are mainly thermal and photo initiation [25,26]. Thermal initiation often attains MIPs with specific surface area, but it is less stable than photo initiation because it affects the interaction between the template and the functional monomer [26]. As seen from Table 1, the most commonly used initiator for flavonoid imprinting was azobisisobutyronitrile (AIBN), which indicated most of the articles used thermal initiation for polymerization. Since flavonoids are naturally active substances and are affected by temperature to a certain extent, it is significant to develop other polymerization initiation methods (e.g., photo initiation) to prepare MIPs in the future. In a word, various methods (mainly bulk polymerization, precipitation polymerization, surface imprinting and emulsion polymerization) have been reported for the synthesis of MIPs towards flavonoids, and scholars are more inclined to study flavonols among all the flavonoids. For a better understanding, these polymerization methods were summarized in Figure 3. Bulk polymerization is the simplest way to synthesize MIPs towards flavonoids, however, this approach requires a large number of templates and grinding steps. Compared with bulk polymerization, precipitation polymerization can generate MIPs with uniform shapes, which greatly improve the use of MIPs for flavonoid separation from natural products. Even so, precipitation polymerization still needs a large amount of solvent and is not environmentally friendly. To solve the problem of difficulty in template elution in both bulk polymerization and precipitation polymerization, surface imprinting has become a common method for synthesis of MIPs towards flavonoids. All in all, researchers can choose the corresponding method to prepare MIPs according to the aim and significance of the study, whereafter applying the prepared MIPs to practical application.

## 3. Applications

MIPs are often used as adsorbents for the separation of flavonoids from different samples because of their specific recognition, fantastic selectivity and tough specificity. In this section, the applications of MIPs in flavonoids are summarized from the following two aspects.

### 3.1. Solid Phase Extraction (SPE)

SPE is one of the most common ways in sample pretreatment [74]. A typical SPE procedure usually includes four steps: (1) activate the SPE column; (2) transfer sample solution to the column; (3) rinse the column with appropriate solvent to remove as much interference as possible; (4) elute the target with solvent (Figure 4). The process of SPE is straightforward to operate, requires less time and is economical in terms of solvent. Recently, the application of MIP-based SPE for extracting flavonoids has also been reported.

#### 3.1.1. MIP Particles Based on SPE

##### Traditional MIP particles Based on SPE

In a traditional SPE method, MIP particles with different sizes are packed into SPE columns. For example, Liang et al. [45] fabricated MIPs ordered microporous MIPs as SPE sorbents to recognize and extract quercetin from Ginkgo Leaves, the recovery of quercetin by the SPE process was up to 55.1%. In addition, isorhamnetin had also been extracted by Li et al. [75] using synthesized specific MIP nanoparticles as SPE adsorbents. Similarly, plentiful MIPs have also been synthesized as SPE adsorbents to selectively extract other flavonoids, such as kaempferol [75], naringin (the recovery was 84.4%) [55], genistein [66], calycosin [28], flavonoid aglycons [28] and myricetin [37].

Although the above tactics have worked well for the target analysis, the traditional MIP-based SPE possessed a tedious step. This method does not provide the advantages of the MIPs compared to the other materials that had no binding specificity.

##### MIP Particles Based on Dispersive Solid-Phase Extraction (dSPE)

Dispersive solid-phase extraction (dSPE) involves the addition of adsorbents to the sample solution, rather than filling them into the SPE column, hence target analytes are adsorbed to the surface of the adsorbents dispersed in the solvent. In this way, the contact area between targets and adsorbents is greatly increased, and the problem of the insufficient contact area of traditional SPE is solved to a certain extent. MIPs towards flavonoids have already been used in SPE. For example, Tong et al. [76] synthesized MIPs towards luteolin by surface imprinting method. Under the optimized conditions of dSPE with the MIPs as adsorbents, extraction of luteolin was successful from four herbs (Feverfew, Senecio Cineraria, Honeysuckle and Semen Plantaginis) and the recovery was 93.9–114%. Similarly, aiming at the separation of kaempferol in Sea-Buckthorn Leaves [77], genistein in milk [78], quercetin in blood samples [31] and red wine [79], MIP-based dSPE in the real samples were also achieved.

Magnetic solid phase extraction is also a kind of dSPE, which mainly uses magnetic or magnetizable materials as adsorption materials. In the dSPE process, there is no need for complicated centrifugal filtration (simplifying the extraction process) (instead of magnetic separation with an external magnetic field) [80,81,82], whereupon relevant magnetic MIPs were fabricated for alternative extraction of flavonoids. Recently, by using a surface imprinting method (with a magnetic Fe_3_O_4_ core), Song et al. [50] synthesized magnetic MIPs towards rutin. By using the magnetic MIPs as dSPE adsorbents, extraction of rutin from Sophora Japonica was successful (the recovery was 87.2–94.6%). Similar essays include: quercetin in onion [38], luteolin in Honeysuckle Leaves [83], metabolites of activated epimedium glycosides in testes and bones of rats [84], baicalein in Scutellaria Baicalensis Georgi [85], kaempferol [52] and quercetin [44] in apples, quercetin, isorhamnetin and kaempferol in Ginkgo Biloba Leaves [86], hesperidin in dried pericarp of Citrus Reticulata Blanco [56] and phloridzin in leaves of Malus doumeri (Bois) A. Chev [60], which were all successfully extracted via magnetic MIP-based dSPE.

Magnetic MIP-based dSPE has been one of the most widely used forms of SPE recently. As seen from the above examples, the prepared magnetic MIPs all showed high binding capacities and high adsorption selectivity towards the corresponding targets. Compared to the traditional SPE, the Magnetic MIP-based dSPE is a preeminent way to prevent packing-related problems as well as the avoiding of the requirement of centrifugation or filtration steps [82,87,88].

#### 3.1.2. Molecularly Imprinted Membranes Based on SPE

Molecularly imprinted membranes (MIMs), which combine the high selectivity of MIPs towards the targets with the physical integrity of the supporting membrane, have also been used as adsorbents in SPE. Recently, Nasir et al. [89] and Turkcan et al. [90] both developed MIMs. The first one prepared a double template MIMs on a polyvinylidene fluoride membrane, while the latter used 2-Hydroxyethyl Metacrylate (HEMA) as constructive monomers and *N*-methacryloyl-(*L*)-histidine (MAH) as functional monomers to develop MIMs with two monomers. The maximum adsorption capacity of MIMs prepared by Nasir et al. [89] was 4.38 mg/L when quercetin was 24.0 mg/L. The maximum adsorption capacity of MIMs synthesized by the latter for quercetin was 299 mg/g, the elution rate was 98.3%, and the adsorption capacity only decreased by about 10.0% after 7 times of repeated use.

Particularly worth mentioning were the MIP fibers designed by Wang et al. [91]. In their work, specific fibers were modified with MIPs (forming MIMs on the surface of the fibers). This solid-phase microextraction method could be classified as a miniaturized sample preparation technique, which would provide a fresh research window for extraction of flavonoids.

It could be seen that the prepared MIMs posed exceptional selectivity and adsorption capacity for the target. Based on the above literature, we believed that we could develop more MIPs for separation of multiple classes of flavonoids with the characteristics of the membrane.

In short, as seen in Table 2, SPE is one of the most common methods for separation of a certain kind of flavonoids. As adsorbents for SPE, MIPs greatly enhance their selectivity, thus attracting more attention in this field. Additionally, the MIPs based on SPE are improving with the innovation of new materials such as magnetic particles, fiber and gel [92] as material.

### 3.2. Sensing

Sensors have the characteristics of high sensitivity, flexibility and high efficiency, which are widely welcomed in the fields of pharmaceutical, biotechnology, environmental testing and industrial contaminant testing [93]. In the literature, MIPs have been successfully used as recognition elements in various sensors (Figure 5) [94,95]. In 1995, Piletsky et al. first achieved sensing herbicides using a MIP-based sensor [94]. Recently, sensing flavonoids using MIP-based sensors has also been reported [96,97,98,99,100,101,102,103,104].

#### 3.2.1. Electrochemical Sensors

Electrochemical sensors are used for quantitative analysis by identifying electrical signals generated by the binding of recognition modules with targets [96]. Electrochemical sensors for the detection of flavonoids have been successfully constructed. Recently, by modifying 3D worm-like nanorod MIPs on glassy carbon electrodes (GCE) with cyclic voltammetry (CV), Meng et al. [97] developed an electrochemical sensor for rutin detection. The LOD for the sensing method was 0.24 × 10^−9^ mol/L. Recently, besides MIPs, other function materials (e.g., MoS_2_ [57], carboxylated multiwalled carbon nanotubes [98] and reduced graphene oxide(rGO) [99,100]) were also modified on the GCE to increase the detection sensitivity. For example, using this modification method, an electrochemical sensor based on Fe_3_O_4_@MIP/rGO/GCE [99] and MoS_2_-MIPs/GCE [57] was developed for sensing luteolin. As anticipated, the LOD was from 0.0400 × 10^−6^ mol/L [57] down to 1.00 × 10^−12^ mol/L [99]. Furthermore, to detect dihydromyricetin from Ampelopsis grossedentata, Hu et al. [100] developed an electrochemical sensor with modified MIPs and rGO on the GCE and the LOD was 1.2 × 10^−8^ mol/L. In addition to GCE-based electrodes, other electrodes can also be combined with MIPs for the detection of flavonoids. For instance, MIPs films were deposited on the surface of indium-tin-oxide (ITO) glass electrode to detect luteolin from traditional medicine Duyiwei capsule. The LOD was 2.4 × 10^−8^ mol/L [58].

Literature shows that MIP-based GCE is widely used as a highly conductive and stable electrode in electrochemical sensors. The introduction of new functional materials can significantly increase the sensitivity of sensing. Therefore, novel electrodes and emerging materials can be introduced as much as possible and applied to detect more flavonoids, in the future.

#### 3.2.2. Optical Sensors

Optical sensors are sensing detectors that measure optical properties in the detection system [101]. Just like the electrochemical sensors mentioned above, there are also surveys on the preparation of optical sensors for the detection of flavonoids by endowing the MIPs with optical response towards the target flavonoids. Recently, construction of optical MIP sensors for detection of flavonoids has been successfully reported [46,47,49,65]. For example, Xu et al. [65] synthesized a fluorescent sensor by integrating graphitic carbon nitride and MIPs, this fluorescent sensor achieved detection of quercetin in grape juice, black tea and red wine. The LOD of this sensing strategy was 2.5 × 10^−3^ mg/L, indicating that the fluorescent MIP sensor was promising in measurements of flavonoids in food. To increase the porosity of the MIP sensor, the luminescent carbon dots were encapsulated into a host of non-luminescent metal-organic frameworks (MOFs) for quercetin sensing [46]. Besides carbon dots, quantum dots were also used as luminescent fluorophores in MIP sensors for detecting flavonoids [47,49]. For example, Li et al. [47] detected cis-diol-containing flavonoids from onion skin and urine using a luminescent sensor that modified with boronate affinity imprinted quantum dots and the LOD of this sensor was 0.20 × 10^−7^ mol/L.

In summary, as seen from Table 3, there are mainly two types of MIPs sensors that have been fabricated for sensing flavonoids. However, only fluorescent MIP sensors have been achieved for detection of flavonoids. Other optical MIP sensors (e.g., optical fiber sensors, colorimetric sensors and Raman scattering sensors) [102] have been never reported. In this case, compared to the optical MIP sensor, the electrochemical MIP sensor is used more frequently. However, the present electrochemical MIP sensors for detection of flavonoids are mainly GCE and ITO electrodes, and there are much more advanced electrodes that can be selected for electrochemical sensing of flavonoids in the future. Moreover, some other MIP-based sensors (e.g., quartz crystal microbalance [103] and Liquefied Petroleum Gas sensors [104]) can also be introduced in the detection of flavonoids.

## 4. Conclusions and Future Perspectives

As an indispensable class of natural products, flavonoids are closely related to human health and diseases. In this review, we summarized the polymerization approaches of MIPs towards flavonoids as template molecules and their application. We found that: (1) Articles on flavonoids mainly focused on quercetin, kaempferol, luteolin and rutin, while MIPs towards morin, chrysin, naringin, myricetin and silymarin were rarely reported. (2) Among the methods of synthesizing MIPs towards flavonoids, surface imprinting and precipitation polymerization were the most commonly used strategies for synthesizing MIPs of flavonoids, emulsion polymerization was a promising way for synthesizing MIPs towards flavonoids. (3) In the case of the application of MIPs for SPE of flavonoids, magnetic MIP-based SPE could be operated simply and quickly. (4) By using MIP sensors to detect flavonoids in actual samples, it was obvious that electrochemical sensors were used more, and only a few optical sensors had been surveyed.

Despite achieving multitudes of accomplishments so far, there still exists some potential challenges and efforts that should be focused on by relevant researchers in molecular imprinting towards flavonoids: (1) Usually, there are many kinds of flavonoids in natural products, and it is a wise choice to isolate a variety of flavonoids from samples simultaneously. At present, the main trend of MIPs preparation towards flavonoids was the production of a single template in most publications. MIPs towards flavonoids using dummy templates or multi-templates should be considered more to achieve separation of multiple flavonoids in future research. (2) Recently, the combination of various technologies with MIPs towards flavonoids is the direction of future scientific development. For example, integration of MIP-based SPE with hollow fiber supported liquid membrane extraction (HF-sLME) could achieve selective extraction of biochanin A in urine [39]. The combination of MIP-based SPE with other methods might exhibit outstanding merits (e.g., enhancing selective extraction ability, effectively facilitating sample analysis and strengthening the ability to connect with instruments preeminently) for extraction of flavonoids from natural products or biological samples. (3) Besides fluorescence MIP sensors for sensing flavonoids, there are also other sensors (e.g., optical stress sensors, fiber-optic hydrogen sensor and Raman scattering sensors could be employed in the detection of flavonoids. (4) Synthesis of electrochemical sensors with novel electrodes (e.g., Carbon ionic liquid electrode, Carbon paste electrode, Carbon nanotubes, Pencil graphic electrode and Carbon fiber paper) for extraction of flavonoids should be exceedingly taken into special consideration. Besides, modification of certain materials on the electrode can also increase its electrical conductivity of the electrode and the sensitivity of detection for flavonoids.

In short, in this review, we provided a summary of the preparation methods (mainly bulk polymerization, precipitation polymerization, surface imprinting and emulsion polymerization) for MIPs towards flavonoids, innovative applications of MIP-based SPE for selective extraction of flavonoids, and sensitive detection for flavonoids by electrochemical sensors and optical sensors. With the development of molecular imprinting technology, sensors and materials science, there is no doubt that more advanced strategies of preparing and applying MIPs with highly specific adsorption capacity will be excavated for extracting flavonoids in the future.

## Figures and Tables

**Figure 1 molecules-27-07355-f001:**
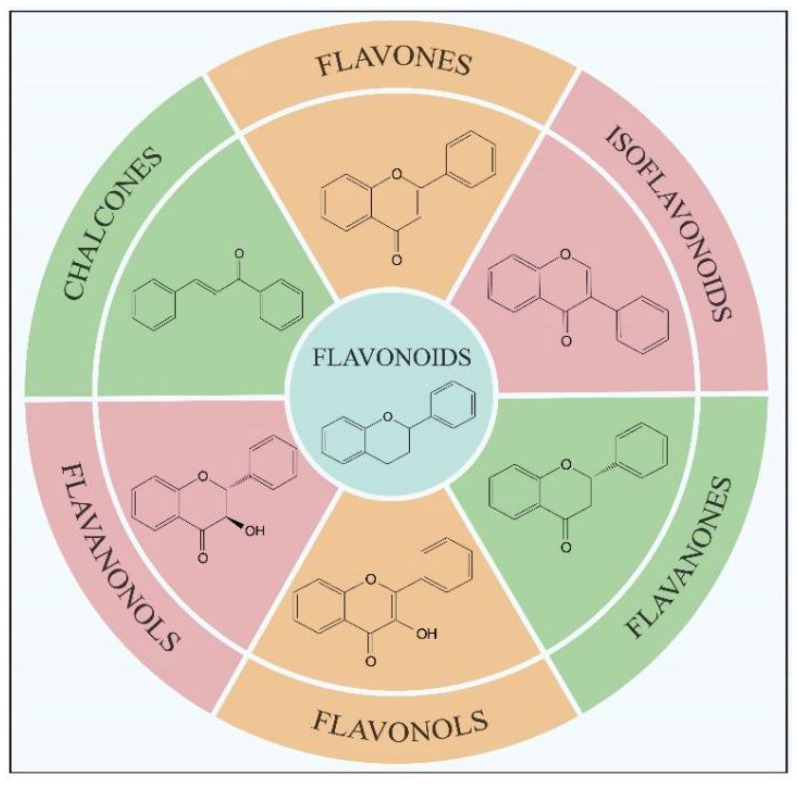
Main subclasses of flavonoids according to their structural formulas.

**Figure 2 molecules-27-07355-f002:**
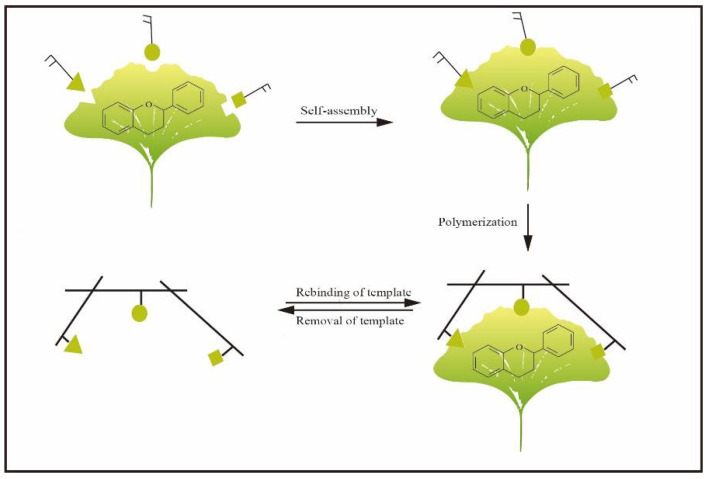
Principle of MIPs preparation.

**Figure 3 molecules-27-07355-f003:**
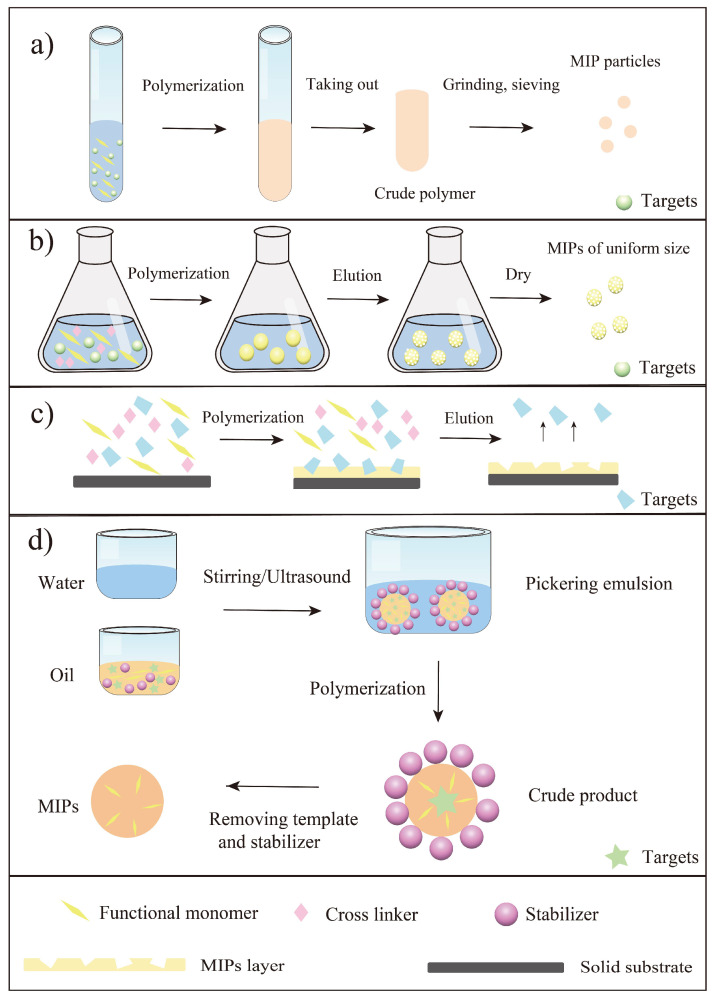
Illustration of main preparation steps of MIPs towards flavonoids using four preparation methods. (**a**) Bulk polymerization; (**b**) Precipitation polymerization; (**c**) Surface imprinting [43]; (**d**) Pickering emulsion polymerization [73].

**Figure 4 molecules-27-07355-f004:**
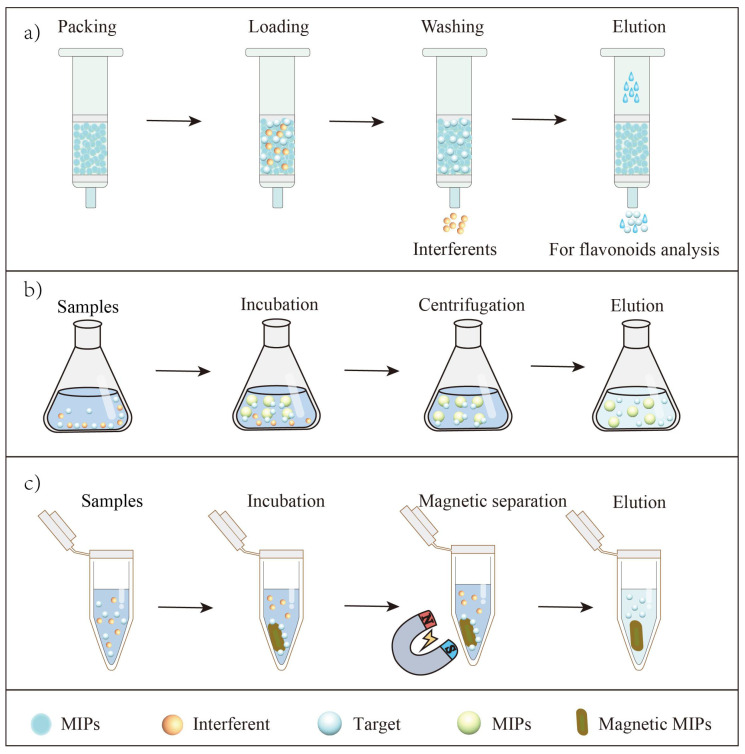
Principles of MIP-based SPE for flavonoid extraction. (**a**) The MIPs towards flavonoids were packed into traditional SPE column for flavonoid extraction; (**b**) The MIPs towards flavonoids were pulled into sample solutions for flavonoid extraction by dispersive SPE; (**c**) The magnetic MIPs were pulled into sample solutions, separating targets and interferents by external magnetic field for flavonoid extraction.

**Figure 5 molecules-27-07355-f005:**
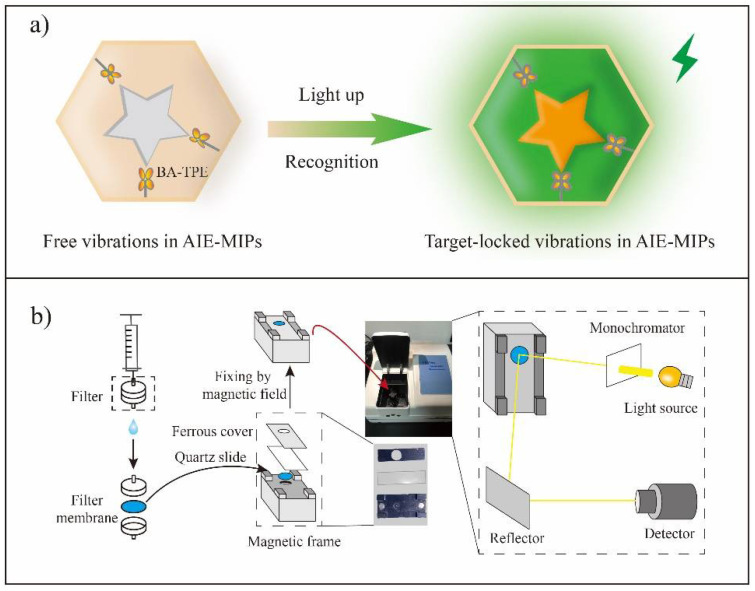
MIP-based sensor. (**a**) The principle of AIE-MIP sensor; (**b**) The illustration of main preparation steps of an AIE-MIP membrane and fixing the AIE-MIP membrane with a magnetic frame for fluorescence detection. This figure is from Yan et al. [95].

**Table 1 molecules-27-07355-t001:** Preparation methods of MIPs towards flavonoids.

Targets Flavonoids	Type of Polymerization	Imprinting System	Polymerization Initiation Methods	Porogen	IF *	Binding Capacity (mg/g)	Ref. *
Flavonoid aglycons	Bulk polymerization	AA *; EDMA *	Thermal initiation	-	-	0.212	[28]
Quercetin		4-VP; EDMA	Thermal initiation	AC *	-	-	[29]
Quercetin		4-VP; EDMA	Thermal initiation	THF *	-	0.397	[30]
Quercetin		DES *; EDMA	Thermal initiation	ACN *	-	23.6	[31]
Schisandrin b		DES; EDMA	Thermal initiation	ACN	-	41.6	[31]
Hesperidin	Precipitation polymerization	AA; EDMA	Thermal initiation	-	2.7	-	[36]
Myricetin		4-VP, GMA *; EDMA	Thermal initiation	MeOH *-ACN	4.9	11.8	[37]
Quercetin		MAA; EDMA	Thermal initiation	MeOH	-	85.5	[38]
Biochanin A		DEM *; EDMA	Thermal initiation	ACN-PhMe *	14	-	[39]
Chrysin		MAA, AA; EDMA	Thermal initiation	-	1.5	210	[40]
Quercetin	Surface imprinting	AA, EDMA	Thermal initiation	-	-	33.0	[43]
Quercetin		MC *-modified Fe_3_O_4_; EDMA	Thermal initiation	ACN-DMSO *	-	10.5	[44]
Quercetin		4-VP; EDMA	Thermal initiation	AC	1.8	-	[45]
Quercetin		4-VP; EDMA	Thermal initiation	-	-	-	[46]
Quercetin Baicalein Luteolin		TEOS *; EDMA	None	-	9.4; 6.6; 11	-	[47]
Quercetin		AA; EDMA	Thermal initiation	-	6.7	0.593 × 10^−4^	[48]
Myricetin		APBA *; EDMA	None	-	7.9	-	[49]
Rutin		DA *, Fe_3_O_4_@HPBA *; EDMA	None	-	7.1	8.10	[50]
Morin		AA; EDMA	Thermal initiation	ACN	3.1	3.20	[51]
Kaempferol		AA; EDMA	Thermal initiation	ACN	3.0	3.84	[52]
Farrerol		4-VP; EDMA	Thermal initiation	-	-	5.80	[53]
Naringin		MAA, AA; EDMA	None	-	-	-	[54]
Naringin		DA; EDMA	None	-	2.9	-	[55]
Hesperidin		*N*-IPAM *; EDMA	Thermal initiation	ACN-MeOH	-	16.6	[56]
Luteolin		EDOT *; EDMA	None	-	-	-	[57]
Luteolin		β-CD *; EDMA	None	-	-	-	[58]
Formononetin		*N*-IPAM, MAA; EDMA	Thermal initiation	-	-	16.4	[59]
Phloridzin		Fe_3_O_4_@SiO_2_@NH_2_; EDMA	Thermal initiation	ACN	3.6	14.6	[60]
Silybin		MAA; EDMA	None	ACN	2.1	15.4	[61]
Quercetin	Emulsion polymerization	TEOS; EDMA	None	-	-	-	[65]
Naringin, genistein		4-VP; EDMA	Thermal initiation	CHCl_3_ *	-	-	[66]
Rutin		DMAPMA *; EDMA	Thermal initiation	DMSO	-	-	[67]
Quercetin		4-VP; DVB *	Thermal initiation	-	4.4	0.521	[73]

* IF: Imprinting factor; Ref: Reference; AA: Acrylamide; EDMA: Ethylene glycol dimethacrylate; 4-VP: 4-vinylphenylboronic acid; AC: Acetone; THF: Tetrahydrofuran; DES: Deep eutectic solvent; ACN: Acetonitrile; GMA: Glycidyl methacylate; MeOH: Methanol; MAA: Methacrylic acid; DEM: 2-(dimethylamino)ethyl methacrylate; PhMe: Toluene; MC: Methacryloyl chloride; DMSO: Dimethyl sulfoxide; TEOS: Tetraethyl orthosilicate; APBA: 3-aminophenylboronic acid; DA: Dopamine; HPBA: Hyperbranched phenylboronic acid; *N*-IPAM: *N*-Isopropylacrylamide; EDOT: 3,4-ethylenedioxythiophene; β-CD: β-cyclodextrin; CHCl_3_: Trichloromethane; DMAPMA: *N,N*-dimethylaminopropyl methacrylamide; DVB: Divinyl benzene.

**Table 2 molecules-27-07355-t002:** Applications of MIPs based on SPE in flavonoids.

Targets	Samples	Separation Method	Recovery (%)	Detection	LOD (μg/mL)	Ref.
Total flavonoids	Astragali Radix	Traditional SPE	97.6	HPLC *-UV *	-	[28]
Myricetin	Safflower flowers of A. manihot		79.8–83.9; 81.5–84.3	HPLC-DAD *	-	[37]
Quercetin	Gingko Leaves		55.1	HPLC-UV	-	[45]
Naringin	Citri Grandis		84.4	HPLC-UV	-	[55]
Naringin Genistein	Herbal medicines		-	HPLC	-	[66]
Quercetin Isorhamnetin Kaempferol	Ginkgo Bloba Leaves		97.6	HPLC-UV	-	[75]
Quercetin Schisandrin b	Blood samples of the mice	dSPE	-	HPLC	-	[31]
Quercetin Schisandrin b	Dried Schisandra, Dried Penthorum		-	HPLC	-	[31]
Luteolin	Four herbs		93.9–114	HPLC	0.020	[76]
Kaempferol	Sea Buckthorn Leaves		>90.0	HPLC	-	[77]
Genistein	Milk		-	MECC *-UV	-	[78]
Quercetin	Red wine		99.7–100	HPLC-UV	0.058	[79]
Quercetin	Red onion	Magnetic dSPE	96.0–98.6	UV	0.06	[38]
Quercetin	Apple		89.2–93.6	HPLC	0.20	[44]
Rutin	Sophora Japonica		87.2–94.6	HPLC	60.0 × 10^−3^	[50]
Kaempferol	Apple		90.5–95.4	HPLC-UV	6.8 × 10^3^	[52]
Hesperitin	Dried Pericarp of Citrus Reticulata Blanco		90.5–96.9	HPLC-DAD	0.60 × 10^2^	[56]
Phloridzin	M. Doumeri Leaves and rats’ Plasma		81.5–90.3	HPLC	0.06; 0.01	[60]
Luteolin	Honeysuckle Leaves		-	HPLC	-	[83]
Activated epimedium glycosides	Bone and testicle of rats		-	UPLC-MS *	-	[84]]
Baicalein	Scutellaria Baicalensis Georgi		91.6–99.3	HPLC-DAD	0.0387	[85]
Quercetin Isorhamnetin Kaempferol	Ginkgo Biloba Leaves		96.8;93.6; 94.8	HPLC-UV	-	[86]
Quercetin	-	MIMs * based SPE	-	-	-	[89]
Quercetin	-		-	-	-	[90]
Hesperetin	Livers of live rats in vivo		81.4–92.9	HPLC	0.02	[91]

* HPLC: High Performance Liquid Chromatography; UV: Ultraviolet ectrophotometer; DAD: diode array detection; MECC: Micellar electrokinetic capillary chromatography; UPLC-MS: Ultra performance liquid chromatography; MIMs: Molecularly imprinted membranes; SPME: Solid-phase microextraction.

**Table 3 molecules-27-07355-t003:** MIP-based sensor for detecting flavonoids.

Targets	Samples	Types of Sensors	Linear Range (mol/L)	Recovery (%)	LOD (mol/L)	Ref.
Luteolin	Gnaphalium Affine	Electrochemical sensor	0.3–30 × 10^−6^	-	0.04 × 10^−6^	[57]
Luteolin	Duyiwei capsule		5.0 × 10^−8^–3.0 × 10^−5^	-	2.4 × 10^−8^	[58]
Baicalein	Baicalein aluminum capsule		0.2–40 × 10^−8^	-	0.6 × 10^−9^	[96]
Rutin	Sophora Japonica		0.05 × 10^−9^–1 × 10^−6^; 0.5–5 × 10^−5^	-	2.4 × 10^−10^	[97]
Genistein	Human urine tablets		0.02–7 × 10^−6^	97.9–103	0.6 × 10^−8^	[98]
Luteolin	Lotus leaves		2.5 × 10^−12^–0.1 × 10^−6^	98.5–101	1.0 × 10^−12^	[99]
Dihydromyricetin	Ampelopsis grossedenta		2.0 × 10^−8^–1.0 × 10^−4^	-	1.2 × 10^−8^	[100]
Quercetin	Ginkgo biloba extract capsule	Optical sensor	0–50 × 10^−6^	-	2.9 × 10^−9^	[46]
Cis-diol-containing flavonoids	Onion skin urine		-	83.5–104; 86.7–105	0.2 × 10^−7^	[47]
Myricetin	Green tea, apple juice		0.3–40 × 10^−6^	-	0.8 × 10^−7^	[49]
Quercetin	Grape Juice tea juice black tea red wine		0.0312–3.1 × 10^−6^	90.7–94.1	2.5 × 10^−9^	[65]
DHF	Mango	Quartz crystal microbalance sensor	-	-	-	[103]

DHF: 2, 5-Dimethyl-4-hydroxy-3(2 H)-furanone.

## Data Availability

Not applicable.

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
