# Peer review of "Preparation and Application of Molecularly Imprinted Polymers for Flavonoids: Review and Perspective"

_molecules, 2022, doi:10.3390/molecules27217355_

Round 1
Reviewer 1 Report
The present manuscript is an interesting review about molecular imprinting technology to detect flavonoids.
The manuscript is well writted and gives a very exhaustive overview of the most relevant works in this particular issue. Moreover, a detailed description of the common polymerization techniques for MIP synthesis are also reviewed.
Tables and figures are adequately presented.
In my opinion, the manuscript should be accepted for publication.
Minor revision:
- Please correct in Figure 3- page 8- "interferents" and "magnetic".
Reviewer 2 Report
This project is certainly interesting because it seizes a lot of opportunities outlined by MIP technology for developing innovative, more stable and easily modulated materials for various applications. The manuscript is informative and from my point of view is suitable for the readership of the Molecules. However, I have several points that could improve the manuscript. An additional one or two figures/tables could be included to make the review easier to catch.
- Line 49: The sentence is deficient: ” Molecular imprinting is a synthetic method for generation of artificial antibodies“… since the MIP are biomimetic materials with a predetermined selectivity for a given analyte or group of structurally related compounds. They can mimic antibodies, enzymes, and other biological molecules; however, in the case of recognition of protein fragments, they are referred to as “plastibodies” (analogy to antibodies) [BelBruno, Joseph J. "Molecularly imprinted polymers." Chemical reviews 119.1 (2018): 94-119; Gao, Mingkun, et al. "Recent advances and future trends in the detection of contaminants by molecularly imprinted polymers in food samples." Frontiers in Chemistry 8 (2020): 616326; Yarman, Aysu, et al. "Simple and robust: The claims of protein sensing by molecularly imprinted polymers." Sensors and Actuators B: Chemical 330 (2021): 129369; Zidarič, Tanja, et al. "Artificial Biomimetic Electrochemical Assemblies." Biosensors 12.1 (2022): 44]. Definitions and concepts in a scientific study must be appropriately used.
- In Section 2 (or as a separate paragraph), some attention should also be paid to materials used as functional monomers.
- Under Section 2.3, entitled Surface imprinting, the authors should also mention MIP synthesis by electropolymerization. In this respect, they could refer to a study by Wei et al. (reference [95] in the manuscript).
- Please make a list of all polymerization initiation methods with their features.
- In Section 3, the authors should spend more words about given applications.
Round 2
Reviewer 2 Report
I would like to compliment the authors for improving the proposed review. They have taken suggestions into consideration and incorporated them into the manuscript.